# The Level of Surface Coverage of Surgical Site Disinfection Depends on the Visibility of the Antiseptic Agent—A Virtual Reality Randomized Controlled Trial

**DOI:** 10.3390/jcm12041472

**Published:** 2023-02-12

**Authors:** Rene Burchard, Lukas Sayn, Ricardo Schmidt, Jan A. Graw, Thomas Scheicher, Christian Soost, Armin Gruenewald

**Affiliations:** 1Department of Orthopedics and Trauma Surgery, University Hospital of Giessen and Marburg, 35043 Marburg, Germany; 2Department of Orthopedics and Trauma Surgery, Lahn-Dill-Kliniken, 35683 Dillenburg, Germany; 3Medical Informatics and Microsystems Engineering, University of Siegen, 57076 Siegen, Germany; 4Department of Anesthesiology and Intensive Care Medicine, Ulm University Hospital, 89081 Ulm, Germany; 5FOM University of Applied Sciences, 45127 Essen, Germany

**Keywords:** surgical site infection, skin antisepsis, remanent disinfectants, knee, virtual reality

## Abstract

Background: Surgical site infections (SSIs) have a significant impact on outcome associated with surgical treatment. Therefore, skin antisepsis has evolved as a standard preoperative procedure in the operating room to reduce the perioperative risk of an SSI. In their “Global Guidelines for the prevention of surgical site infections”, the World Health Organization (WHO) recommend the use of an agent with remanent additives and considers colored agents as helpful. However, colored and remanent disinfectants are not available in Germany. The aim of the present study was to investigate whether using a colored antiseptic solution increases the quality of preoperative skin antisepsis. Methods: This study was designed as a randomized, double-blinded controlled trial. To examine the level of coverage of skin antisepsis, an appropriate virtual reality (VR) environment was generated. Participants could see a movable surgical clamp with a swab in their hand. When touching the skin, the participants recognized an optical change in the appearance of the skin: Using a colored antiseptic solution resulted in orange-colored skin. Using an uncolored agent, a shiny wet look was visible without a change in natural skin color. Results: Data of 141 participants (female: 61.0% (*n* = 86); mean age: 28 y (Range 18–58 y, SD = 7.53 y)) were included in the study. The level of disinfection coverage was higher in the group using the colored disinfectant. On average, 86.5% (sd = 10.0) of the leg skin was covered when a colored disinfectant was used, whereas only 73.9% (sd = 12.8) of the leg skin was covered when the participants had to use an uncolored agent (*p* < 0.001, effect size: *f* = 0.56, *η*^2^ = 0.24). Conclusions: The use of an uncolored disinfectant leads to a lower surface coverage of the perioperative skin disinfection. Thus far, it is unclear whether using uncolored disinfectants is associated with higher risks for perioperative infections compared with the use of non-remanent disinfectants. Therefore, further research is necessary and current German guidelines should be re-evaluated accordingly.

## 1. Introduction

Surgical site infections (SSIs) have a significant impact on outcome associated with surgical treatment and can be considered as a relevant burden for patients, surgeons, and the health care system [1]. When infection control was introduced by Ignaz Semmelweis in the 19th century, the patient’s skin was identified as one of the major sources of the pathogens for an SSI [2]. Therefore, skin antisepsis has evolved to a standard preoperative procedure in the operating room to reduce the perioperative risk of an SSI [3].

Different compositions of skin disinfectants based on alcoholic or aqueous solutions, including various additional agents, are currently available. The Guidelines Development Group (GDG) of the World Health Organization (WHO) recommends the use of alcohol-based antiseptic solutions with an additional remanent agent [4]. Currently available remanent additives are chlorhexidine-gluconate (CHG) or povidone-iodine (PVP-I). WHO board members prefer CHG agents, based on several available studies such as the work of Darouiche and colleagues, who stated that CHG was superior to PVP-I additives for preoperative skin cleaning during clean-contaminated surgery [5]. Similar results were found in the context of cesarean sections [6]. In Germany, guidelines of the Robert Koch Institute follow the WHO recommendations [7].

Besides choosing the best agent for the most effective skin antisepsis, colored antiseptic solutions were introduced with the aim to visualize appropriate skin coverage by the skin disinfectant [8]. The effect of a colored skin preparation was studied with regard to the patients’ skin pigmentation [8]. McDaniel and co-authors showed videos of accurately and non-accurately prepared forearms to orthopedic surgeons that evaluated the adequacy of the skin preparation. Based on their results, they recommend different colored antiseptic agents for dark- and fair-skinned patients. However, the results do not support the general use of colored agents. Thus, the use of a colored and remanent disinfection agent for perioperative disinfection appears optimal. However, colored disinfectants containing CHG are not approved in Germany, and are therefore not available [7]. This poses a problem for hospitals and surgeons in selecting a suitable agent.

Therefore, the aim of the present study was to investigate whether a colored antiseptic solution can increase the level of coverage of the preoperative skin preparation in comparison to an uncolored antiseptic solution.

## 2. Materials and Methods

### 2.1. Study Design

The study was conducted as a randomized double-blinded controlled trial (RCT) in 2020. In total, 146 students (female: 61.0% (*n* = 89); mean age: 28 y (Range 18–58 y, SD = 7.46 y)) from one University of Applied Sciences were recruited. Exclusively, students with no medical backgrounds were selected to prevent experiential bias regarding disinfection procedures in the operating room. All students were randomized to the study arms. After data collection, outlier analysis was performed, and 141 students remained in the study. The outlier analysis included 5 study participants whose disinfection coverage was unusually low; thus, it was assumed that these participants had problems handling the VR environment. The CONSORT^®^ participant flow diagram is shown in Figure 1. The randomization ensured an equivalent pattern of epidemiologic parameters in both groups. Investigators were blinded for group assignment until the end of the experiment. Written informed consent was obtained from all participants. The study was approved by the Ethical Review Board Westphalia-Lippe Medical Association, Muenster, Germany (No.: 2020-033-f-S), and registered as a clinical trial (German Clinical Trials Register: DRKS00027522).

### 2.2. Test Setting

To examine the level of coverage of skin antisepsis, an appropriate virtual reality (VR) environment was generated with the Unity-Engine V2019.3.4f1, Unity Software Inc., San Francisco, CA, USA. In this environment, a typical setting of an operation room with a fair-skinned simulation patient in supine position and a partial draped leg mounted in a leg holder was modelled using Blender 2.83 Software (Blender 2.83, Blender Foundation, Amsterdam, The Netherlands). Two leg manifestations were generated: a lean leg and a plump leg. The study participants were provided with VR glasses and a VR controller (HTC Vive Pro, HTC Corporation, New Taipei, Taiwan/Valve Corporation, Bellevue, WA, USA). In the presented VR environment, participants could see a typical surgical clamp with a swab in their hand and were able to move this tool in the virtual room. When they were touching the skin of the patient’s leg, the study participants could recognize an optical change in the appearance of the skin. In one group, the coloring led on to an orange-colored leg; in the other group, a shiny wet look was visible without a change in the natural skin color. The allocations of the leg characteristics and the disinfectant used were distributed randomly.

The surface of the leg was split into segments with an average size of 1.6 cm × 1.6 cm each. The lean leg had a surface area of 0.27 m^2^ covered by approximately 1000 segments; the plump leg had a surface area of 0.41 m^2^ with approximately 1400 segments. The overall coloring of all segments was analyzed at the end of the procedure. Figure 2 shows the view for both cases (Figure 2a: orange-colored disinfectant solution; Figure 2b: uncolored disinfectant solution). Participants were able to move in the OR, to crouch down, and thus look at the leg from all sides. There was no timeout in the trial, and pre-trial training of handling the VR tools was performed. The VR modelling was supervised by a board-certified senior orthopedic surgeon who conducted reality checks for each environment setting during the complete process.

### 2.3. Statistical Analysis

Pre-trial power analysis was performed using G*Power. A mean effect size according to McDaniel et al., an alpha-error of 5%, and a minimum power of 80% according to Cohen was assumed [8,9]. The result was a minimum sample of 128 participants, which was exceeded here with a sample of 141 participants [10]. The statistical analysis was calculated with R 4.0.3 (open source). After a logit transformation of the dependent variable, a two-way independent ANOVA with interaction effect was performed [11]. Levene’s test showed variance homogeneity between groups (*p* > 0.05 for all groups); thus, a robust F-statistic could be assumed.

## 3. Results

All 141 study participants were able to fully complete their trial run. There was neither a technical drop-out nor participants based during the test process. Table 1 shows the characteristics of the study participants. A balanced distribution into the different study arms was ensured: 74 (52.5%) participants were allocated in the group using colored disinfectant solution, whereas 67 (47.5%) participants used clear disinfectant solution (*p* = 0.614). Furthermore, 66 (46.8%) participants handled a lean leg, while 75 (53.2%) participants handled a plump leg (*p* = 0.501).

The level of coverage of disinfection coverage as the primary outcome parameter was higher in the group using colored disinfectant solution (Figure 3a). On average, 86.5% (sd = 10.0) of the leg skin was covered using a colored disinfectant, whereas only 73.9% (sd = 12.8) of the leg skin was covered when the participants had to use an uncolored agent (*p* < 0.001, effect size: *f* = 0.56, *η*^2^ = 0.24) (Figure 3a). In contrast, the size of the leg had no significant impact on the percentage of the covered skin area (82% (sd = 11.4) for a lean leg vs. 79.2% (sd = 14.2) for a plump leg, *p* = 0.414, effect size: *f* = 0.07, *η*^2^ = 0.004) (Figure 3b).

## 4. Discussion

Using a virtual reality setting, the aim of the present study was to investigate whether coloring a skin disinfectant increased the level of coverage of surgical site preparation. The results show that the use of a colored agent improves the coverage of the skin preparation before surgery.

The WHO considers “coloring agents” as “helpful to indicate where surgical site preparation products have been applied on the patient’s skin” in their “Global Guidelines for the prevention of surgical site infections” [4]. However, to date, data are scarce which support this expert opinion. Nevertheless, surgical site preparation has been an intensively debated topic in the literature [5,6,8,12,13,14,15,16,17,18,19]. Most studies and analyses compare outcome and complication rates between different antiseptic agents, with a particular interest in the comparison of aqueous and alcohol-based solutions. Several studies claim a lower risk for suffering from an SSI when using disinfectants based on chlorhexidine-alcohol compared with povidone-iodine solutions or alcohol without chlorhexidine after total joint replacement, cesarean delivery, abdominal surgery, or urologic procedures [5,6,12,13,14,15,20]. Some authors considered the use of octenidine-alcohol as the most effective skin antisepsis [16]. Others could not show any advantages or disadvantages when they compared chlorhexidine with povidone-iodine as skin preparation agents [17,18]. Davies and Patel combined the two compounds, suggesting a lower risk of SSI in comparison to the individual use of one of them [19]. Taken together, there is conflicting evidence regarding the use of chlorhexidine-alcohol-, povidone-iodine-, or solely alcohol-based solutions for skin antisepsis. However, in Germany, the Robert Koch Institute recommends the use of a disinfectant with the addition of a remanent active ingredient [7].

To the best of our knowledge, only McDaniel et al. addressed the relevance of using colored skin disinfectants for surgical site preparation [8]. The authors analyzed whether the visibility of a colored disinfectant differed by various skin pigmentations. However, based on their results, this working group only concluded that the use of agents adapted to the patients’ skin color was beneficial. Although they recommended the use of colored agents in general, this was not the subject of their study.

Despite the significant difference between the two study cohorts, even in the colored disinfectant group, incomplete skin coverage of 13.5% was noted. All study participants were non-professionals without a clinical background, which could partly explain the lack of experience. However, the significant difference between both study groups is nonetheless striking. Thus, further research regarding the level of skin coverage achieved by experienced surgeons seems sensible.

To enroll a randomized controlled prospective study on the topic of effectiveness of coloring disinfectants during the COVID-19 pandemic in an environment of contact restrictions and the temporary shut-down of clinical study activities, a VR approach was used. VR is a computer technology enabling the user to fully immerse themselves in a digital world by wearing a head-mounted display. Immersion into VR yields the perception of being physically present in a virtual environment, which has become increasingly realistic due to significant advances in technology in recent years [21,22]. In addition, presence in VR also refers to being cognitively in the virtual world, e.g., by having plausible interactions with the virtual environment. There is robust evidence that when two aspects of VR are present, i.e., the illusion to be in a virtual place as well as the plausible behavior inside the virtual environment, people act how they would in the real world [23]. For this reason, VR is used in various fields of scientific research (data visualization), entertainment (VR games and movies), and education and training (remote teaching and simulations) [24]. It has been proven to be a successful tool in medical training applications such as for laparoscopic surgery; it can be seen that VR training improves the efficiency and quality in trainees’ surgical practice, as well as tissue handling [25]. Several other successful applications of VR as a simulation tool in the medical field have already been presented, such as VR CT simulation and virtual reality and simulation in neurosurgical training [26,27].

The virtual environment developed for this study simulated a typical operating room setting, including the presence of a surgical table, operating lights, and a patient prepared for knee surgery. To increase the user’s sense of presence, the user could move around freely inside the operating room and the interaction between the surgical clamp and the leg was modelled to be as realistic as possible by using a spring mechanism, which prevented the swab from vanishing in the virtual leg after contact. Great care was also taken in modeling the color and light reflections on the virtual leg to be as close to reality as possible. Therefore, impressions from a real OR setting were used to model the light conditions of the virtual environment. The color of the disinfectant and the size of the leg were randomized automatically by the VR software.

Nevertheless, the present study has some limitations. The use of the VR environment is a very modern and new scientific approach; therefore, there validation data about clinical techniques in the operation room are scant. After definition as a clinical trial in retrospect, this VR study was registered retrospectively with the German Clinical Trials Register. Furthermore, complete randomization failed to achieve equal gender distribution, with an overall majority of female study participants. However, to date, there are neither data describing a gender effect on the level of coverage of skin preparation before surgery, nor any evidence that gender influences the outcome in contemporary VR studies [28]. The study participants had not received any medical education, and therefore had no experience in preoperative skin preparation. However, the study participants were therefore not biased regarding specific previously known antiseptic agents and techniques for surgical site preparation. Furthermore, the study design allowed no prediction for the risk to suffer from an SSI after skin disinfection with either detergent. In addition to the possibility of conducting this RCT in a COVID-19-restricted environment, the VR setting enabled virtual clinical research to be performed without exposing any patient at risk to any study-associated adverse effects.

## 5. Conclusions

The use of an uncolored disinfectant leads on to a lower surface coverage of skin disinfection. Thus far, it is unclear whether using uncolored disinfectants is associated with higher risks for perioperative infections compared with the use of non-remanent disinfectants. Nevertheless, further clinical research is necessary to confirm the results of this VR-based study and to investigate the effect regarding the rate of SSIs.

## Figures and Tables

**Figure 1 jcm-12-01472-f001:**
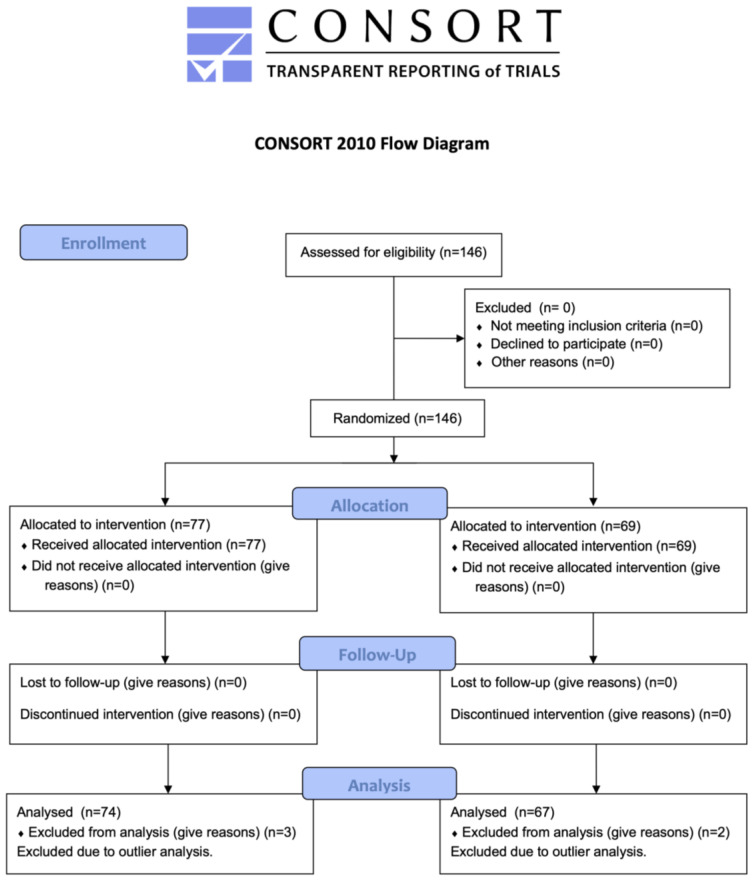
CONSORT^®^ flow diagram.

**Figure 2 jcm-12-01472-f002:**
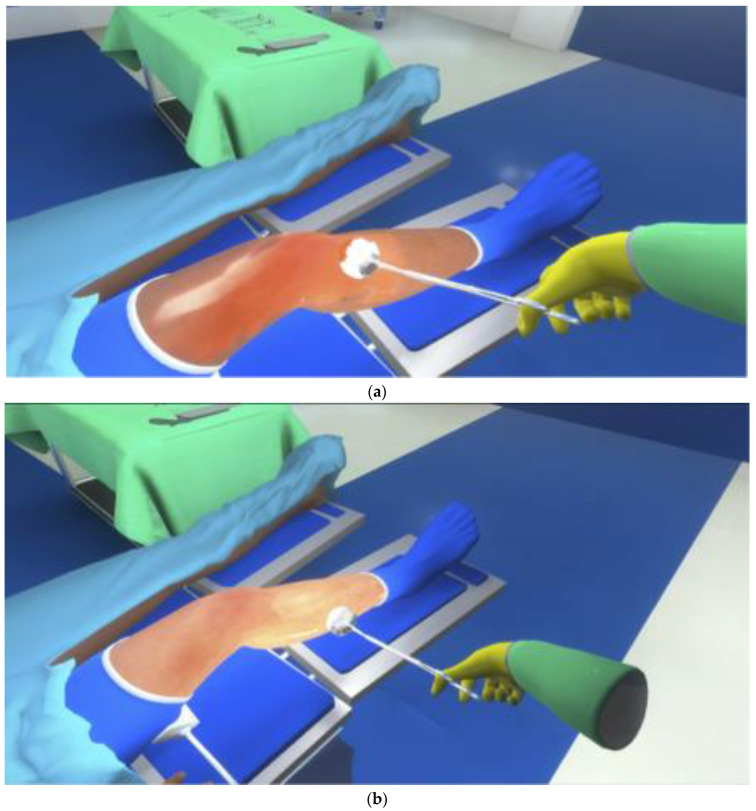
(**a**) Representative screenshot of the VR simulation: skin antisepsis in a virtual operating room using a surgical clamp with a swab and (orange) colored disinfectant solution. (**b**) Representative screenshot of the VR simulation: skin antisepsis in a virtual operating room using a surgical clamp with a swab and uncolored (clear) disinfectant solution.

**Figure 3 jcm-12-01472-f003:**
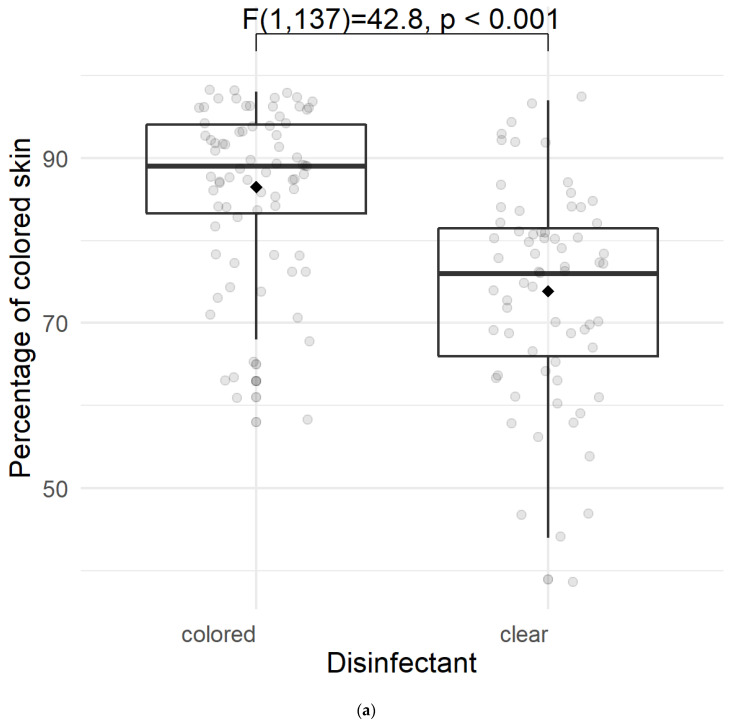
(**a**) Impact of using a colored vs. an uncolored disinfectant solution on the level of coverage of skin disinfection. (**b**) Impact of anatomical leg dimensions on the level of coverage of skin disinfection.

**Table 1 jcm-12-01472-t001:** Epidemiologic parameters of the study participants.

	Colored (N = 74)	Clear (N = 67)	Total (N = 141)	*p* Value
Gender				0.002
male	20 (27.0%)	35 (52.2%)	55 (39.0%)	
female	54 (73.0%)	32 (47.8%)	86 (61.0%)	
VR Experience				0.470
no	59 (79.7%)	50 (74.6%)	109 (77.3%)	
yes	15 (20.3%)	17 (25.4%)	32 (22.7%)	
Handedness				0.920
right-handed	72 (97.3%)	65 (97.0%)	137 (97.2%)	
left-handed	2 (2.7%)	2 (3.0%)	4 (2.8%)	
Leg				0.425
lean	37 (50.0%)	29 (43.3%)	66 (46.8%)	
plump	37 (50.0%)	38 (56.7%)	75 (53.2%)	
Age				0.886
Mean (SD)	28.24 (7.67)	28.06 (7.44)	28.16 (7.53)	
Range	20–58	18–55	18–58	

## Data Availability

The data presented in this study are available on request from the corresponding author. The data are not publicly available due to ethical board request.

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
