# Peer review of "The Level of Surface Coverage of Surgical Site Disinfection Depends on the Visibility of the Antiseptic Agent—A Virtual Reality Randomized Controlled Trial"

_jcm, 2023, doi:10.3390/jcm12041472_

Round 1
Reviewer 1 Report
Title
Level of surface coverage of surgical site disinfection depends on the visibility of the antiseptic agent – a virtual reality randomized controlled trial
Authors
Rene Burchard * , Lukas Sayn , Ricardo Schmidt , Jan Adriaan Graw , Thomas Scheicher , Christian Soost , Armin Gruenewald
Thank you for a well written manuscript. This manuscript was really interesting reading and felt very innovative and informative. The manuscript fits the journal and is clinical relevant. I think you have a clear message and it is easy to follow throughout the manuscript.
I have only one questions for you to ponder.
I have prepared many patients throughout the years and I have also used both clear and colored skin disinfection solutions. The method used for skin disinfection containing ethanol is based on applying the solution several times and for a certain time frame. My experience is that coverage is not an issue, but your VR research results says it is. This conflict is kind of interesting. My question is in regards to coverage when applying skin disinfection several times and with experienced staff. What are your thoughts on the applicability of your research results towards the clinical working settings? I would feel it would be important for your results to be discussed in regards to my question.
But all together a very clear and concise study!
Kind regards,
Author Response
Thank you very much for this important comment. During the VR setting the study participants were able to perform skin disinfection for several times. They aimed to cover all areas of the presented leg until they felt that complete and appropriate disinfection was applied. We believe that the VR environment is sufficiently mimicking a clinical reality in the clinical environment and the development process of the VR environment was continuously accompanied by experienced surgeons to provide the most realistic experience possible. However, we fully agree with the reviewer that the inexperience of the subjects could reduce the transferability of the results to the clinical setting. We have therefore formulated the limited general transferability of the VR setting to the operating room in the limitations section of the revised version of the manuscript: lines 216-218 (highlighted in yellow) and we have included additional limitations of a potential interoperator bias on lines 223-225 of the revised version of the manuscript (highlighted in yellow).
Reviewer 2 Report
The manuscript by Rene Burchard et al. describes a virutal reality study to evaluate a simple hygienic parameter, the sanitized surface during surgical management.
The manuscript suffers from a few items in need of revision.
Line 335: Level of evidence?
Line 37/39: delete.
Methods : Give information about outlier analysis and about randomization (as the consort indicate desequilibrated randomization).
Method: How was the number of subjects to be added in addition to the 128 determined?
put "vs." in italics.
The legends of the boxplot have been inverted.
Author Response
Line 335: Level of evidence?
Line 35 was deleted according to the Reviewer´s concern.
Line 37/39: delete.
Line 37/39 was deleted according to the Reviewer´s suggestion.
Methods : Give information about outlier analysis and about randomization (as the consort indicate disequilibrated randomization).
We decided to perform an outlier analysis in both disinfection groups, taking out very small values for the disinfected surfaces (2 in the transparent disinfectant group, 3 in the colored disinfectant group), as these very small values indicate incorrect use by the subjects in the VR environment. When the downward outliers are included in the analysis, the results remain unaffected and change only marginally in the coefficients. Accordingly, the results are robust. We have now described the outlier analysis in more detail in the revised manuscript (lines 81-83).
Randomization was performed by a computerized random number generator. In addition, participants had to randomly distribute themselves between two rooms. We see no evidence for unbalanced randomization even if the significant difference in gender frequency might suggest it. Previous studies on disinfectant use have not demonstrated a gender difference. Nevertheless, we pointed out this fact in the limitations (lines 230-232). All other sociodemographic characteristics did not show significant differences between the groups.
Method: How was the number of subjects to be added in addition to the 128 determined?
Since we expected a higher dropout or outliers, the sample size was deliberately set higher than the required minimum number of participants of 128 to ensure a sufficient sample size in any case. Subsequently, it did not seem reasonable to us to reduce the number of participants to 128 if 141 participants could be evaluated.
put "vs." in italics.
“vs.” was put in italics according to the Reviewer´s suggestion.
The legends of the boxplot have been inverted.
We thank the reviewer for locate these errors. Therefore, we revised the boxplot legends in the manuscript.